# Incidence, recurring admissions and mortality of severe bacterial infections and sepsis over a 22-year period in the population-based HUNT study

**Kristin Vardheim Liyanarachi**[1,2]*, **Erik Solligård**[1,3], **Randi Marie Mohus**[1,3], **Bjørn O. Åsvold**[4,5,6], **Tormod Rogne**[1,3,7☯], **Jan Kristian Damås**[1,2,8☯]

**1** Gemini Center for Sepsis Research, Department of Circulation and Medical Imaging, NTNU, Norwegian University of Science and Technology, Trondheim, Norway, **2** Department of Infectious Diseases, St Olavs Hospital, Trondheim University Hospital, Trondheim, Norway, **3** Clinic of Anaesthesia and Intensive Care, St Olavs Hospital, Trondheim University Hospital, Trondheim, Norway, **4** Department of Endocrinology, Clinic of Medicine, St Olavs Hospital, Trondheim University Hospital, Trondheim, Norway, **5** K.G. Jebsen Center for Genetic Epidemiology, Department of Public Health and Nursing, NTNU, Norwegian University of Science and Technology, Trondheim, Norway, **6** HUNT Research Center, Department of Public Health and Nursing, NTNU, Norwegian University of Science and Technology, Levanger, Norway, **7** Department of Chronic Disease Epidemiology and Center for Perinatal, Pediatric and Environmental Epidemiology, Yale School of Public Health, New Haven, Connecticut, United Ststes of America, **8** Centre of Molecular Inflammation Research, Department of Clinical and Molecular Medicine, NTNU, Norwegian University of Science and Technology, Trondheim, Norway

☯ These authors contributed equally to this work.
* kristin.v.liyanarachi@ntnu.no

**Data Availability Statement:** Our data cannot be shared publicly due to patient confidentiality. Data

## Abstract

### Purpose

Severe bacterial infections are important causes of hospitalization and loss of health worldwide. In this study we aim to characterize the total burden, recurrence and severity of bacterial infections in the general population during a 22-year period.

### Methods

We investigated hospitalizations due to bacterial infection from eight different foci in the prospective population-based Trøndelag Health Study (the HUNT Study), where all inhabitants aged ≥ 20 in a Norwegian county were invited to participate. Enrollment was between 1995 and 1997, and between 2006 and 2008, and follow-up ended in February 2017. All hospitalizations, positive blood cultures, emigrations and deaths in the follow-up period were captured through registry linkage.

### Results

A total of 79,393 (69.5% and 54.1% of the invited population) people were included, of which 42,237 (53%) were women and mean age was 48.5 years. There were 37,298 hospitalizations due to infection, affecting 15,496 (22% of all included) individuals. The median time of follow-up was 20 years (25th percentile 9.5–75th percentile 20.8). Pneumonia and urinary tract infections were the two dominating foci with incidence rates of 639 and 550 per

from the HUNT Study used in research projects is available upon request to the HUNT Data Access Committee (hunt@medisin.ntnu.no) to research groups who meet the data availability requirements (described here: http://www.ntnu.edu/hunt/data).

**Funding:** This study was supported by Samarbeidsorganet Helse Midt-Norge, NTNU Norwegian University of Science and Technology (Trondheim, Norway) (KL). The funders had no role in study design, data collection and analysis, decision to publish, or preparation of the manuscript. The funder provided support in the form of salaries for authors [KL, ES, RMM, BOÅ, TR, JKD], but did not have any additional role in the study design, data collection and analysis, decision to publish, or preparation of the manuscript. The specific roles of these authors are articulated in the 'author contributions' section.

**Competing interests:** The authors have declared that no competing interests exists.

100,000 per year, respectively, and with increasing incidence with age. The proportion of recurring admissions ranged from 10.0% (central nervous system) to 30.0% (pneumonia), whilst the proportion with a positive blood culture ranged from 4.7% (skin- and soft tissue infection) to 40.9% (central nervous system). The 30-day mortality varied between 3.2% (skin- and soft tissue infection) and 20.8% (endocarditis).

## Conclusions

In this population-based cohort, we observed a great variation in the incidence, positive blood culture rate, recurrence and mortality between common infectious diseases. These results may help guide policy to reduce the infectious disease burden in the population.

## Introduction

Severe bacterial infections are common causes of hospital admission and are associated with adverse outcomes such as sepsis and death [1–4]. Bacterial infections and sepsis are substantial and increasing problems worldwide [5], and The World Health Organization (WHO) has called for initiatives aimed at increasing knowledge that can contribute to a reduced burden of sepsis.

In any disease process, understanding the epidemiology is mandatory as background information when deciding which measures to implement and which resources to prioritize. Many previous studies have information on the incidence rates and mortality of infections [2, 6–8], however, the population burden of infectious diseases rely on additional factors. In particular, risk of recurrence and risk of systemic infection are often not assessed.

In this paper we describe the burden of hospitalization for groups of bacterial infections in a large Norwegian population-based cohort of 79,393 patients followed over a 22-year period. In addition to describing the incidence rates, we have estimated the 30-day all-cause mortality, the proportion of recurring admissions, and the positive blood culture rate within the different infection foci.

## Materials and methods

### Description of the study cohort

The Trøndelag Health Study (HUNT Study) is a series of cross-sectional surveys conducted in Nord-Trøndelag from 1984 where all inhabitants aged 20 years or older were invited to participate [9]. The Nord-Trøndelag region in central Norway has a population of approximately 130,000. It consists of rural areas and small towns and is considered generally representative of Norway with regard to sources of income, age distribution, morbidity and mortality, but the average income and prevalence of higher education and current smoking are a little lower than the Norwegian average [10].

We used data from the second and third surveys, HUNT2 (1995–1997) and HUNT3 (2006–2008), respectively, in which a total of 79,393 subjects agreed to participate (69.5% and 54.1% of the invited population for HUNT2 and HUNT3). The majority of the participants (72% of the women and 69% of the men) in HUNT2 also participated in HUNT3. The participants completed questionnaires covering a wide range of health-related topics, underwent clinical examination and blood collection and were then followed from the day of first inclusion and up until February 2017. For all participants, we retrieved information on all hospital admissions to the county hospitals or the regional tertiary care hospital.

## Classification of infectious diseases

The International Classification of Disease (ICD) by WHO is the foundation for the identification of health trends and statistics globally [11] and is used in many countries, including Norway, for administrative/economic purposes upon hospital discharge. The 10th revision (ICD-10) is currently being used, prior to 1999 the codes were from the 9th revision (ICD-9). We identified all ICD-9/10- codes describing a potentially serious bacterial infection and categorized them into 8 main groups: pneumonia, UTI (urinary tract infection), SSTI (skin- and soft tissue infection), IAI (intraabdominal infection), CNS (central nervous system) infection, endocarditis, bone- and joint infection and sepsis/bacteraemia. Each admission during the study period with one of the infection-codes was then identified and grouped (S1 Table).

A widely used method to identify patients with sepsis, is to use a combination of primary sepsis codes (explicit sepsis) and codes for infection with a known organ focus combined with codes for organ dysfunction (implicit sepsis) [12–16]. In 2020, Rudd et al [2] published global, regional and national sepsis incidence and mortality data using this type of approach. In our study, patients identified as having had sepsis as defined by their criteria were used in the further discussion on sepsis.

## Study design and statistical analyses

We retrieved the ICD-9 and ICD-10 codes for all hospitalizations of the study subjects in the county hospitals and to the regional tertiary care hospital. All Norwegian citizens are assigned a unique identification number at birth, and this number is registered in health care contacts. In addition to accessing the ICD codes upon discharge, this identification number was used to link data from the HUNT Study with the Norwegian population registry to obtain information on date of emigration and date of death, as well as to the hospitals´ information on positive blood cultures through February 2017.

Incidence rate was defined as incidence per 100,000 person-years of a first-time infection. A recurring admission was defined as a new admission with the same infection occurring more than 30 days after the first admission, and the proportion having a recurring admission was calculated among patients who survived the first 30 days after the first admission. Mortality was defined as death within 30 days of admission of a last-time infection. A first-time infection with a positive blood culture was defined as a blood culture being positive within 30 days of the admission. The blood cultures were taken on clinical indication only. Isolates commonly associated with skin contamination were not considered (e.g.coagulase-negative Staphylococci). We performed the estimations on all sites of infections, however, chose to focus on pneumonias, UTIs and SSTIs. All analyses were carried out using StataMP version 16.

## Ethics approval

The project has been approved by the Regional Committee for Medical and Health Research Ethics of Central Norway, REK Midt (2006/393-4), (2009/1717-2), (2014/144), (2016/55). All participants signed an informed consent before entering the HUNT study.

## Results

From the date of HUNT entry and up until February 2017, 15,496 (22%) of the 79,393 participants were hospitalized due to a bacterial infection at least once. Background characteristics of our study populations are described in Table 1.

**Table 1. Background characteristics of the HUNT2 and HUNT3 population.**

|  | All (N = 79,393) | HUNT2 (N = 65,665) | HUNT3 and not HUNT2 (N = 13,728) |
|---|---|---|---|
| Age on participation (years) | 46.8 (34.4–61.9) | 48.9 (36.4–64.3) | 37.1 (27.3–48.9) |
| Time followed (years) | 20.0 (9.5–20.8) | 20.2 (15.3–20.9) | 9.2 (8.8–9.7) |
| Male sex | 37,156 (46.8) | 30,710 (46.8) | 6442 (46.9) |
| Died during follow-up | 19,539 (24.6) | 19,002 (28.9) | 533 (3.9) |
| Emigrated during follow-up | 353 (0.44) | 246 (0.37) | 107 (0.78) |

Data are presented as n (%) for dichotomous characteristics and median (25th percentile-75% percentile) for continuous characteristics.

The median follow-up-time was 20.0 years (25th percentile 9.5–75th percentile 20.8). The total number of hospital admissions with an infection (first-time and recurring events) was 37,298 (Table 2). 4628 patients had two infection-codes during the same admission, and 981 had three or more. The most common hospitalizations were due to pneumonia, UTIs and sepsis/bacteraemia. There was also a substantial number of intrabdominal infections and SSTIs. There was a small number of CNS infections, endocarditis and bone-/joint infections (Fig 1).

## Pneumonias

A total of 7,948 people had a first-time pneumonia, and the incidence rate of 639 per 100,000 per year increased with age from 106 at the age of 30 to 3,200 after the age of 80 (Fig 2). Of them 4.9% had a positive blood culture. Recurring admissions with pneumonia were frequent (Table 2), as 30% of the survivors of a first-time pneumonia had a subsequent readmission with the same diagnosis. Of them 54.8% occurred within the first year, and 96.6% within the first ten years. Of the patients admitted with a first-time pneumonia, 15.6% died during or after this admission or of a subsequent pneumonia. Pneumonia was the focus of infection which seemed to have the largest seasonal difference, with a markedly higher incidence rate in September- February compared to the warmer months. (S2 Table).

## Urinary tract infections

A total of 6,839 participants had an admission with a first-time UTI, giving an incidence rate of 550 per 100,000. This again increased steeply with age from 90 at the age of 30 to 2473 after the age of 80, and there was a marked difference between men and women, with women

**Table 2. Summary of results divided into eight different foci of infection.**

| Focus of infection | Total admissions (n) | First-time admissions (n) | Incidence rate pr 100 000/year (95% CI) | Proportion with recurrent infection (%, 95% CI) | Proportion with positive blood culture (%, 95% CI) | 30-day mortality (%, 95% CI) |
|---|---|---|---|---|---|---|
| Pneumonia | 13,210 | 7,948 | 639 (625–653) | 30.0 (28.9–31.0) | 4.9 (4.5–5.5) | 15.6 (14.8–16.4) |
| UTI | 11,421 | 6,839 | 550 (537–563) | 28.6 (27.5–29.7) | 7.5 (6.9–8.1) | 8.5 (7.8–9.1) |
| Sepsis/bacteraemia | 6,956 | 4,156 | 334 (324–345) | 20.9 (19.6–22.2) | 37.5 (36.1–39.0) | 13.3 (12.3–14.4) |
| IAI | 2,904 | 2,008 | 161 (154–169) | 11.6 (10.3–13.1) | 9.8 (8.6–11.1) | 3.6 (2.9–4.5) |
| SSTI | 2,204 | 1,429 | 115 (109–121) | 16.1 (14.2–18.1) | 4.7 (3.8–6.0) | 3.2 (2.4–4.3) |
| Bone/joint | 247 | 173 | 14.0 (12.0–16.2) | 12.7 (8.5–18.6) | 19.1 (13.9–25.7) | 7.0 (4.0–11.9) |
| Endocarditis | 238 | 96 | 7.7 (6.3–9.4) | 22.2 (14.4–32.7) | 39.6 (30.2–49.8) | 20.8 (13.8–30.2) |
| CNS | 118 | 65 | 5.2 (4.1–6.7) | 10.0 (4.5–20.8) | 40.9 (29.6–53.3) | 7.7 (3.2–17.4) |

UTI, urinary tract infection; IAI, intraabdominal infection; SSTI, skin- and soft tissue infection

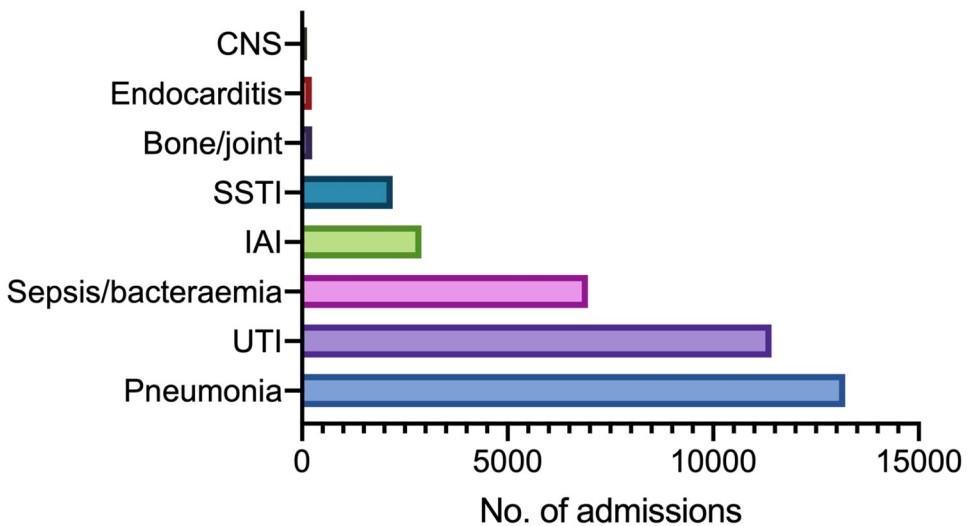

**Fig 1. Number of admissions by foci of infection.** Distribution of 37,298 admissions with different foci.

having an incidence rate of 637 per 100,000 (S3 and S4 Tables). Of all the first-time UTIs 7.5% had a positive blood culture. The readmission rate for UTI was 28.6%. Of them 55.1% occurred within the first year, and 96.6% within the first ten years. 8.5% of the patients admitted with UTI, died during or after this admission or of a subsequent UTI (Fig 2).

## Skin-/soft tissue infections

A total of 1,429 participants had a first-time SSTI, giving an incidence rate of 115 per 100,000 and this again increased significantly with age. 16.1% of patients surviving a SSTI had a readmission. In this group, the number of readmissions had a profound variation, up to 43 readmissions were registered in some patients. Of them 58.2% occurred within the first year, and 98.0% within ten years. Of the first-time SSTIs 4.7% had a positive blood culture and 3.2% of the patients admitted with a SSTI died during or after this admission or of a subsequent SSTI (Fig 2).

## Sepsis

Amongst our 37,298 admissions with a bacterial infection, 3,687 fulfilled the criteria for explicit sepsis and 1,864 for implicit sepsis, making the total number of admissions being counted as caused by sepsis 5,224 (14.1%) (as defined by Rudd et al [2]). This constituted 12.2% of the pneumonias, 7,6% of the UTIs and 7.4% of the SSTIs.

3,671 admissions fulfilled the criteria for a first-time sepsis, giving an incidence rate of 295 per 100,000, increasing with age, and 1,499 (40.8%) had a positive blood culture.

The readmission rate was 12.9%. Of them, 48.3% occurred within the first year and 55.6% within 10 years. Of the patients admitted with sepsis 12.9% died during or after this admission or of a subsequent sepsis (Fig 2).

## Discussion

In this prospective study of ~80,000 subjects representative of the adult Norwegian population, we observed different patterns of incidence, recurrence, blood culture positivity and mortality within the different foci of bacterial infection.

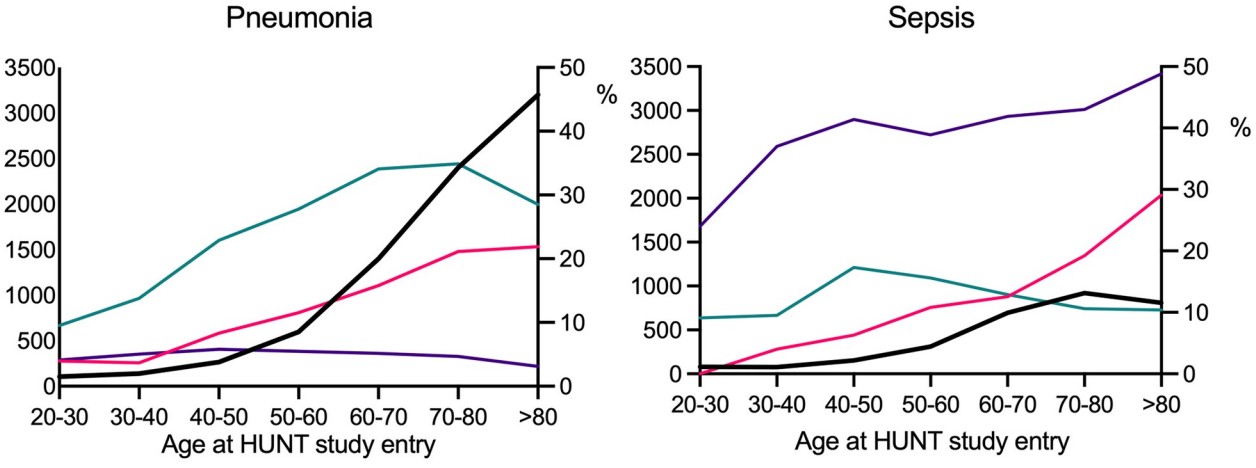

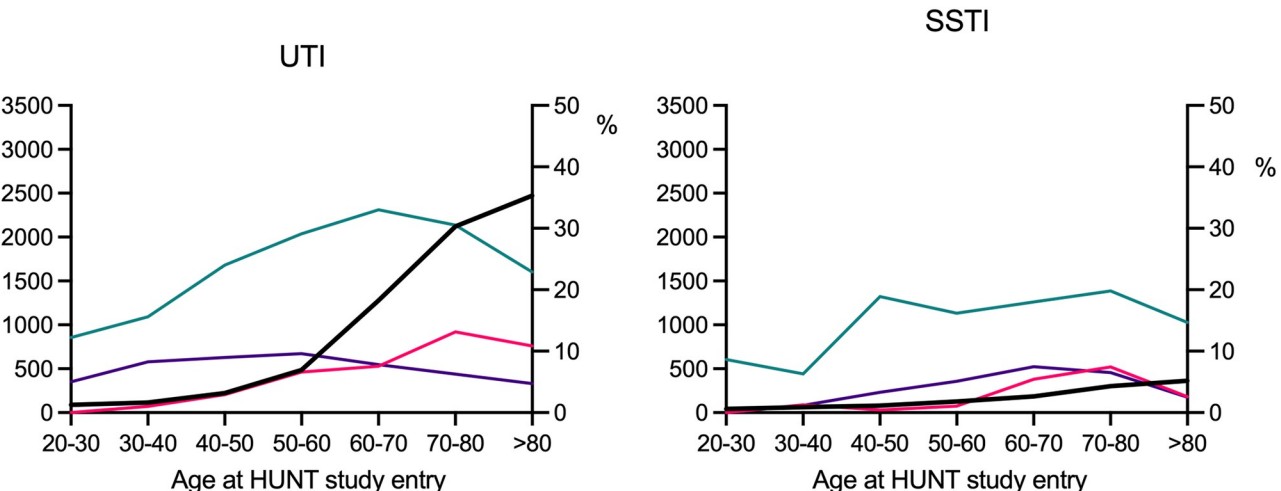

**Fig 2. Incidence rates, rate of recurrent infections, 30-day-mortality rate and rate of positive blood cultures by age group.** The distribution of incidence rates (black line), recurrence (green line), 30-day mortality (red line) and proportion of positive blood cultures (purple line) in the 3 chosen foci of infection and in sepsis as defined by Rudd et al. On the x-axis is the age upon entering the HUNT study. The left y-axis shows incidence rate per 100,000 per year, the right y-axis shows proportions (in %) of recurrence, 30-day mortality and positive blood cultures. UTI, urinary tract infection; SSTI, skin- and soft tissue infection.

## Incidence

Pneumonia was the dominating focus of infection, with the highest incidence rate, followed by the UTIs. The distribution of the different foci of infection is comparable and correlates with earlier work based on discharge codes in a hospital´s catchment area [8, 17, 18]. The incidence rate of hospitalization due to pneumonia correlates well with a calculation of incidence rate in an emergency setting in New York [7], however one should be careful when comparing our results with numbers in an emergency room or intensive care setting. A Danish study from 2006 [6] reports incidence rates that are lower than we have found, however their main conclusion was that pneumonia incidence is on the rise. They report an increase in hospitalized pneumonia from 288 to 442 per 100 000 person-years from 1994 to 2003, and we found the rate to be 639 up to 14 years later.

The incidence rates of all the different groups of infections, including the participants having an admission qualifying as sepsis, increased steeply with increasing age. In 84% of all admissions the patient was 60 years or older, and in 39% the patient was above the age of 80. This is also the case for blood stream infections (a positive blood culture), as concluded by Mehl et al in 2017 in analyses within our study region [19]. Our long follow-up time which naturally led to our participants being followed into their old years combined with the fact that most of the infections appeared in the elderly population, probably is the explanation why as many as 22% of the study participants had a hospital stay during the follow-up-time.

There are several factors possibly explaining why the elderly are predisposed to infections [20, 21]. Frailty and functional limitations, comorbidities leading to immunosuppression and polypharmacy all play important roles both in first-time and recurring cases. Milbrandt et al reviewed the epidemiology of critical illness in the elderly in 2009, describing the ageing and less responsive immune system, the higher risk of nosocomial infections and the higher risk of sepsis [22]. With an ageing population, the burden of severe bacterial infections will likely further increase, both with respect to the number of admissions and disease severity. In addition to ageing, the role of other underlying conditions and modifiable risk factors is certainly interesting factors in future research.

## 30-day-mortality-rate

Our mortality rates were based on the last infection of each focus. This has necessarily given a higher case-fatality rate compared to having the first-time infections as the denominator, however, this describes deaths from both the first-time infections and the recurrences, and we believe it has given a more correct description of the total burden. In the case of co-infection it was not possible to determine which of the foci of infection contributed the most to death. This could, in theory, have over-estimated the mortality rate of especially UTI, which is a commen "co-infecter".

Our estimated mortality rate due to pneumonia is in line with a previous study [23]. Rudd et al found that globally, both sexes and all age groups combined, the most common underlying cause of sepsis-related death was pneumonia every year from 1990 to 2017 [2].

In our cohort, the incidence of pneumonia and the 30-day mortality rate from pneumonia increased with increasing age. Death from pneumonia have previously been linked to frailty. Kundi et al showed in 2019 that patients with higher frailty scores had higher observed rates of 30-day postadmission mortality and 30-day post discharge mortality in addition to a higher 30-day readmission rate [24]. For the SSTIs the mortality rate is low. Hardly any studies have reported on the overall mortality rates for SSTIs, however Kaye and colleagues found that mortality was relatively low in the United States and decreased from 0.56% in 2005 to 0.46% in 2011 [25].

Our 30-day all-cause mortality for sepsis was 12.9% and the 30-day mortality rate increased with age. The all-cause mortality was lower than others studies have found [8]. In the global estimations performed by Rudd et al in 2020, the sepsis mortality varied substantially across regions, being greatest in sub-Saharan Africa, Oceania, south Asia, east Asia, and southeast Asia. They did not calculate the total 30-day mortality rate from sepsis, however the age-standardized percentage of all global deaths related to sepsis was 20.1% [2].

### Recurring infections

Our large pre-defined patient cohort gave us the opportunity to look further into the recurrent admissions. However, our rates of recurring infections and the finding that a large proportion of the readmissions are within the first year, are not easily comparable to others. Most other have counted readmission rates within 30 days [26–28], this being more a proxy for treatment failure or premature discharge. In addition, recurring infection of all causes is most often counted. This is especially common in USA, where 30-day-all-cause-readmission rate of pneumonia is being closely monitored as part of a "Hospital Readmissions Reduction Program» [24, 26].

In the group with SSTI, the rate of admission due to recurring infection was stable throughout the age group, however there was a huge variation in the number of readmissions, from 1 to 43. Recurring infections have been shown to be a major contributor to the overall SSTI burden [29], and in our population there seems to be a subgroup of patients with frequent recurring admissions. This group should be studied further, for example by assessing if genetic predisposition could play a role [30].

### Positive blood cultures/bacteremia

Overall we found a low rate of positive blood cultures. This is not surprising, as we know that bacterial infections are often managed without identifying the causative microorganism [31, 32]. This is particularly true with pneumonia, however, patients with pneumonia and bacteremia are shown to have a high in-patient mortality rate [33].

The rate of 7.5% of positive blood cultures in UTI were substantially lower than described by Artero et al in 2016 [34], however in this study the patients were identified by having clinical features of UTI and not by discharge codes. As opposed to pneumonia, they then conclude that the presence or absence of bacteremia in elderly people with UTI requiring hospitalization does not influence in-hospital mortality. The low positive blood culture rate in SSTIs (< 5%) was expected and matches previous findings [25]. In this group of infection, the clinicians must, more than in other conditions, make their treatment decision without knowing the causative pathogen. Due to this low rate of blood culture positivity, blood cultures are in fact not routinely recommended in the American practice guidelines [35].

Of the first-time admissions qualifying as explicit or implicit sepsis 40.8% had a positive blood culture. This fits in with the existing knowledge that identifying a patient with bloodstream infection will identify many but not the majority of sepsis patients. In the work performed by Nygård et al, 37% of sepsis patients defined by clinical criteria had a positive blood culture [31]. It is found than in an intensive care setting, 40% of patient defined as having "severe sepsis" do not have findings in their blood cultures [32].

### ICD coding

The method of retrospectively identifying infections using ICD codes is not without challenges. Henriksen et al found that using ICD codes in this way will underestimate the true burden, however it has a high degree of validity when stratifying on the different sites of infection

[36–38]. Skull et al found that using ICD-10 codes to identify pneumonias was a valid method, even likely to be superior to the use of symptoms and signs or interpretation of radiology reports [39]. The first important challenge is the possibility of change in coding practice, regulations, guidelines and tradition over time, especially when the observation period is as long as 22 years. Another is the variation within each of the organ-groups when it comes to precision and practical usage of the different codes. Some codes describe a well-defined disease entity whilst others describe a whole spectrum of disease. This variation was the reason why we chose not to look closer at the IAIs, whilst on the other hand focused on the SSTIs. Pneumonias and UTIs were chosen because they were the two dominating groups, and they are, in our experience, fairly precisely coded.

The correct identification of patients with sepsis is a particular challenge. Iwashyna et al concluded that using ICD codes is "reasonable" compared to going through medical records [12], whilst others have highlighted marked discrepancies [38, 40, 41]. Gaieski et al have pointed out that even within the studies retrospectively using ICD codes, the different methods of databank abstraction will give a substantial variation [42]. Counting only the primary sepsis codes will make the sepsis definition too narrow and lead to a gross underestimation, as doctors mostly are guided to only use these codes when the focus of infections is unknown [43]. When it comes to using the criteria defined by Rudd et al, critics have claimed that their definition of sepsis is too broad and that sepsis is likely to be over diagnosed. Especially the process of choosing which infectious and non-infections conditions should count toward the entity "implicit sepsis" is complicated [44]. There is ongoing work with finding better and more refined definitions of sepsis based on ICD coding.

## Strengths

The major strength of our study includes its large size, its long-term follow-up, as well as the linkage to microbiological records from all the laboratories at the local and regional hospitals. A few population cohorts have earlier looked at bacterial infections, however a median follow-up time of a 20 years, the opportunity to look at recurring events, the linkage to the positive blood cultures and lastly the fact that we look at all the different foci of bacterial infection, gives us a unique insight into the total burden of disease in this population comparable to the rest of the western world.

Sepsis is difficult to identify not only clinically but also retrospectively. Identifying bacterial infections based on the focus of the infection/infected organ and not sepsis directly, seems be a more correct way of getting av overview of the true burden of disease as this fit more in with how the coding practice actually works.

## Limitations

Our long-time follow up is a strength, but also brings the challenge of having to consider changes within the healthcare system, including changes in health-seeking behaviour in a more informed cohort over time.

As always in this kind of prospective study, there is a possibility of selection bias. 54–69% of the invited population participated, and although this is a high participation rate for a health survey, different known high-risk groups for infection, such as intravenous drug users, will probably be underrepresented.

In addition, the question of how representative this population is of the general population is increasingly important, as more and more people live in cities where the risk factors of living in urban areas will have to be taken into account. We have compared our results with earlier

results from different countries, however an overweight of comparable studies were from USA. It is uncertain how comparable our cohort is to an American population.

Outpatient antibiotic use prior to hospitalization could clearly have influenced the low level of positive blood cultures on most foci of infection. Unfortunately, our data does not give us the possibility to assess this further.

## Conclusions

In this paper we describe the burden of severe bacterial infections in a large Norwegian population cohort during a 22-year follow up by presenting the incidence rates, 30-day mortality and proportions of positive blood cultures and recurring infections with particular focus on pneumonias, UTIs, SSTIs and sepsis. It points out the substantial number of recurring infections within all infection foci, especially in the SSTIs and it points out that most severe bacterial infections will not be identified through blood cultures, not even the admissions qualifying as "sepsis" by the internationally used sepsis criteria. Our data clearly describe the substantial number of hospitalizations due to severe bacterial infections and how both the number of admission and severity of disease most likely will increase in the future, seeing the clear increase of both incidence and mortality with increasing age.

## Supporting information

**S1 Table. Overview of the different ICD-9 and ICD-10 codes selected to identify the bacterial infections.**
(PDF)

**S2 Table. Seasonal differences in incidence rates.**
(PDF)

**S3 Table. Summary of results divided into eight different foci of infection, males.**
(PDF)

**S4 Table. Summary of results divided into eight different foci of infection, females.**
(PDF)

## Acknowledgments

The Trøndelag Health Study is a collaboration between the Trøndelag Health Study Research Centre (Faculty of Medicine and Health Sciences, Norwegian University of Science and Technology), the Trøndelag County Council, the Central Norway Regional Health Authority, and the Norwegian Institute of Public Health.

## Author Contributions

**Conceptualization:** Kristin Vardheim Liyanarachi, Erik Solligård, Jan Kristian Damås.

**Data curation:** Kristin Vardheim Liyanarachi, Tormod Rogne.

**Formal analysis:** Kristin Vardheim Liyanarachi, Tormod Rogne.

**Funding acquisition:** Erik Solligård, Jan Kristian Damås.

**Investigation:** Kristin Vardheim Liyanarachi.

**Methodology:** Kristin Vardheim Liyanarachi, Randi Marie Mohus, Bjørn O. Åsvold, Tormod Rogne, Jan Kristian Damås.

**Project administration:** Erik Solligård, Jan Kristian Damås.

**Resources:** Kristin Vardheim Liyanarachi, Erik Solligård, Jan Kristian Damås.

**Software:** Kristin Vardheim Liyanarachi.

**Supervision:** Tormod Rogne, Jan Kristian Damås.

**Writing – original draft:** Kristin Vardheim Liyanarachi.

**Writing – review & editing:** Kristin Vardheim Liyanarachi, Erik Solligård, Randi Marie Mohus, Bjørn O. Åsvold, Tormod Rogne, Jan Kristian Damås.

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
