## [Decision Letter · Decision Letter 0]

2 Mar 2022

PONE-D-21-20080

Incidence, recurring admissions and mortality of severe bacterial infections and sepsis over a 22-year period in the population-based HUNT Study.

PLOS ONE

Dear Dr. Liyanarachi,

Thank you for submitting your manuscript to PLOS ONE. After careful consideration, we feel that it has merit but does not fully meet PLOS ONE’s publication criteria as it currently stands. Therefore, we invite you to submit a revised version of the manuscript that addresses the points raised during the review process.

The reviewers response were positive. Please follow all of the advice given.

We look forward to receiving your revised manuscript.

Kind regards,

Kazumichi Fujioka

Academic Editor

PLOS ONE

Journal Requirements:

2. Thank you for stating the following financial disclosure: "This study was supported by Samarbeidsorganet Helse Midt-Norge, NTNU Norwegian University of Science and Technology (Trondheim, Norway) (KL). The funders had no role in study design, data collection and analysis, decision to publish, or preparation of the manuscript."

We note that one or more of the authors is affiliated with the funding organization, indicating the funder may have had some role in the design, data collection, analysis or preparation of your manuscript for publication; in other words, the funder played an indirect role through the participation of the co-authors. If the funding organization did not play a role in the study design, data collection and analysis, decision to publish, or preparation of the manuscript and only provided financial support in the form of authors' salaries and/or research materials, please do the following:

a. Review your statements relating to the author contributions, and ensure you have specifically and accurately indicated the role(s) that these authors had in your study. These amendments should be made in the online form.

b. Confirm in your cover letter that you agree with the following statement, and we will change the online submission form on your behalf: 

“The funder provided support in the form of salaries for authors [insert relevant initials], but did not have any additional role in the study design, data collection and analysis, decision to publish, or preparation of the manuscript. The specific roles of these authors are articulated in the ‘author contributions’ section.

Reviewers' comments:

Reviewer's Responses to Questions

**Comments to the Author**

1. Is the manuscript technically sound, and do the data support the conclusions?

Reviewer #1: Yes

Reviewer #2: No

2. Has the statistical analysis been performed appropriately and rigorously? 

Reviewer #1: Yes

Reviewer #2: No

3. Have the authors made all data underlying the findings in their manuscript fully available?

Reviewer #1: Yes

Reviewer #2: Yes

4. Is the manuscript presented in an intelligible fashion and written in standard English?

Reviewer #1: Yes

Reviewer #2: No

5. Review Comments to the Author

Reviewer #1: The manuscript is well written, with relevant informative figures/table. Its mainly a descriptive and observational study - so statistical analysis is limited to baseline comparisons.

Language is very good.

Reviewer #2: This study gives a population-based overview of common severe infections and sepsis in a Norwegian area with a background population of 130,000. The study is mainly descriptive.

I have some major concerns that confuse me:

It is based on patients that have agreed to participate in a study, based on questionnaires, clinical examination, and retrieval of blood collection. However, this is mentioned once and then all the data used in the study are registry-based. Does the legislation in Norway not allow retrieval of registry-based data without the consent of each individual patient? If so, this should be stated explicitly.

What is meant by “blood collection”? Does that include both biochemistry and microbiological specimens? If it also includes the latter, did that have any impact on the rate of positive blood cultures (i.e., not taken on indication only)?

The authors claim that this study is prospective. In what way is it prospective, especially given that data from the questionnaires apparently were not reported in this study? It seems to be a historic registry-based study.

The time frames are confusing. The study mentions a 22-y time period, but later we find out that this includes two separate cohorts, one from 1995-1997, the other from 2006-8. Moreover, the expression “22-y follow up” is used frequently. The follow-up period differs between the two cohorts of which the oldest has around 11 years more to get recurrent infections etc. I cannot figure out where a 22-y follow-up period comes from. A more appropriate used of follow-up would e.g. be for the 30-d mortality rate, i.e. that is a 30-day follow-up period. A study period is not the same as a follow-up period.

An example: “From the date of HUNT entry and up until February 2017, the 79,393 participants had 37,298 hospital admissions with a bacterial infection (first-time and recurring event) (Fig 1).”: These participants were found in two cohorts (how many in each?), of which the first had 11 years more follow-up time. And if the age distribution was the same in the two cohort, those from the first cohort were 11 years older at the same calendar time, which has a huge impact on incidence etc. So these data are blurred and muddled.

Nothing is mentioned about differences and similarities between the two cohorts. Were there any changes in incidence, mortality etc.??

It is difficult for non-Scandinavians to know how data from different registries were merged, as most countries do not have a unique personal identification number for their citizens.

Although the authors briefly describe the difficulties of defining sepsis, including the coding of these, they have omitted some important studies that show that the incidence of sepsis, based on ICD-codes, may vary more than three-fold [1,2]. These reviews were, amongst others, based on three studies [3-5], one of which was from Norway, but even this study is not mentioned in this manuscript [4].

I am not an expert on all the infections described in the article, but concerning pneumomia there are studies that give a thorough overview, are population-based, and have a very high number of patients [6].

It is confusing that some analyses are based on the first-time and some on the last time occurrence. The baseline should be the same (first-time occurrence) and competing-risk analyses [7] should be incorporated.

“To the best of our knowledge, this is the first time such a large population cohort have been studied with such a long follow-up, a population comparable to the population of the rest of Norway/western world”: I tend to disagree, as mentioned above for pneumonia, and much larger background populations than 130,000 have also been the basis for studies of e.g. bacteremia [8].

Minor comments:

In the abstract is written: “Thorough background information on the total burden and severity of the different foci of infection will contribute to reduce this.”: How will some information per se reduce some burden and severity of different foci of infection??

Too many results from the tables and figures are reiterated in the text in the “Results” section.

References

1. Wilhelms SB, Huss FR, Granath G, et al. Assessment of incidence of severe sepsis in Sweden using different ways of abstracting International Classification of Diseases codes: difficulties with methods and interpretation of results. Critical care medicine. 2010 Jun;38(6):1442-9.

2. Gaieski DF, Edwards JM, Kallan MJ, et al. Benchmarking the incidence and mortality of severe sepsis in the United States. Critical care medicine. 2013 May;41(5):1167-74.

3. Angus DC, Linde-Zwirble WT, Lidicker J, et al. Epidemiology of severe sepsis in the United States: analysis of incidence, outcome, and associated costs of care. Critical care medicine. 2001 Jul;29(7):1303-10.

4. Flaatten H. Epidemiology of sepsis in Norway in 1999. Crit Care. 2004 Aug;8(4):R180-4.

5. Martin GS, Mannino DM, Eaton S, et al. The epidemiology of sepsis in the United States from 1979 through 2000. The New England journal of medicine. 2003 Apr 17;348(16):1546-54.

6. Thomsen RW, Riis A, Nørgaard M, et al. Rising incidence and persistently high mortality of hospitalized pneumonia: a 10-year population-based study in Denmark. J Intern Med. 2006 Apr;259(4):410-7.

7. Fine JP, Gray RJ. A proportional hazards model for the subdistribution of a competing risk. Journal of the American Statistical Association. 1999;94:496-509.

8. Nielsen SL, Lassen AT, Gradel KO, et al. Bacteremia is associated with excess long-term mortality: A 12-year population-based cohort study. J Infect. 2015 Sep 9;70(2).

6. PLOS authors have the option to publish the peer review history of their article (what does this mean?). If published, this will include your full peer review and any attached files.

Reviewer #1: **Yes: **Harriet Mayanja-Kizza

Reviewer #2: **Yes: **Kim Oren Gradel

---

## [Author Response · Author response to Decision Letter 0]

7 Apr 2022

Dear Editor,

We are excited about the opportunity to revise and resubmit our manuscript “Incidence, recurring admissions and mortality of severe bacterial infections and sepsis over a 22-year period in the population-based HUNT Study”. 

We found the comments from the reviewers highly relevant and addressing the issues raised has improved the manuscript. 

The reviewers’ comments are included in this resubmission, with our point-by-point response.

The revised manuscript, one marked and one unmarked version, is enclosed. 

2b:The text regarding funding can be changed according to your advice: “The funder provided support in the form of salaries for authors [KL, ES, RMM, BOÅ, TR, JKD], but did not have any additional role in the study design, data collection and analysis, decision to publish, or preparation of the manuscript. The specific roles of these authors are articulated in the ‘author contributions’ section.

3: Please change our Data Availability statement to reflect this correct information: Our data cannot be shared publicly due to patient confidentiality. Data from the HUNT Study used in research projects is available upon request to the HUNT Data Access Committee (hunt@medisin.ntnu.no) to research groups who meet the data availability requirements (described here: http://www.ntnu.edu/hunt/data)

Sincerely,

Kristin Vardheim Liyanarachi. 

Comments from the reviewers:

Reviewer 1: 

This is an interesting well designed cohort study looking at population severe bacterial infection, with outcomes of determining specific infection site incidence rates, 30 days’ all-cause mortality, recurring admissions, positive blood culture. The cohorts were enrolled in 1995-1997 (70% acceptance rate) and 2006-2008– (54% acceptance rate). It would be of interest to determine if they were outcome differences between the two cohorts regarding severe infection over the years

Comment 1

Was there any overlap in the HUNT 2 and 3 cohorts? What was retention like over the years.

Response: Yes, there was a substantial overlap, and this has been clarified in the revised manuscript. 72% of the women and 69% of the men that joined HUNT 2 joined HUNT 3 ten years later. The date of entry to our study was set as the day these people joined HUNT 2. They were not counted twice. The retention was high. Only 353 people (0.4% of the study population) emigrated out of the study area during the follow-up-time. 19.539 people (24.6%) died. A table with background characteristics of the two HUNT cohorts has now been included as Table 1 (line 134) and the following changes has been made in the revised manuscript on page 6, lines 84-91: 

“We used data from the second and third surveys, HUNT2 (1995-1997) and HUNT3 (2006-2008), respectively, in which a total of 79,393 subjects agreed to participate (69.5% and 54.1% of the invited population for HUNT2 and HUNT3). The majority of the participants (72% of the women and 69% of the men) in HUNT2 also participated in HUNT3. The participants completed questionnaires covering a wide range of health-related topics, underwent clinical examination and blood collection and were then followed from the day of first inclusion and up until February 2017. For all participants, we retrieved information on all hospital admissions to the county hospitals or the regional tertiary care hospital.”

Comment 2 

The study was conducted in what can referred to as “a rural area”. However, urban areas may have various factors impacting on risks of infections and their outcomes, such as environmental factors – outdoor vs indoor lifestyles etc – should these be considered in the limitations.

Response: This is a very good point made by the reviewer. An increasing number of people live in cities of increasing sizes also in this region of Norway. Fortunately, the next cohort in this ongoing study (HUNT4), has increased its catchment area and now includes Trondheim, Norway´s third largest city. We have now included this important point in the limitations of our study on page 17, lines 333-337: 

“In addition, the question of how representative this population is of the general population is increasingly important, as more and more people live in cities where the risk factors of living in urban areas will have to be taken into account.”

Comment 3 

Regarding the “implicit sepsis” related to organ dysfunction – could they have been an over diagnosis of sepsis in this group. 

Response: As correctly observed by the reviewer this could most certainly be the case, and this is an important aspect of discussion when using the entity implicit sepsis as a way of retrospectively identifying sepsis. We have broadened the discussion regarding this in the revised manuscript on page 16, lines 309-311:

“When it comes to using the criteria defined by Rudd et al, critics have claimed that their definition of sepsis is too broad and that sepsis is likely to be over diagnosed. Especially the process of choosing which infectious and non-infections conditions should count toward the entity “implicit sepsis” is complicated [46].”

Comment 4 

The low blood culture noted is an important finding – as clinicians often assume this is due to prior antibiotics. Was this possibility of prior outpatient antibiotic use explored?

Response: This is again a very important aspect of our study raised by the reviewer, and could clearly have influenced on our data. However, our data does not give us the possibility to assess this further as we do not have access to information on outpatient antibiotic use prior to hospitalization. This important aspect has now been included as a limitation of our study on page 17, lines 338-340: 

“Outpatient antibiotic use prior to hospitalization could clearly have influenced the low level of positive blood cultures on most foci of infection. Unfortunately, our data does not give us the possibility to assess this further.”

Comment 5 

About half the population got at least one infection needing admission – is this a true reflection of this Norwegian population– or due to higher cohort awareness? It is quite high especially as the age inclusion included young adults. In the questionnaire were participants advised to report to hospital for any suspected infection, or this is an expected figure in a none cohort population of a mixed age population from 20 yrs up upwards. What was effect of age on admissions? 

Response: Of the total of 37,298 admissions with infection, 15,496 was a first-time admission. We see that this could appear unclear, as we also use the terms “first-time pneumonia”, first-time UTI” etc. This means that approximately 22% of the patients had one admission or more due to infections in our population. Although increased cohort awareness could have contributed to increased identification of infections, the decision to admit the patients to hospital for infections is in most cases made by the general practitioner. Accordingly, increased cohort awareness may increase numbers of diagnosed infections all over, but should not have a major impact on the high numbers of hospitalized patients as reported in our study. As suggested by the reviewer, it could also be partly explained by the long- term follow-up studying participants into their old years. In fact, most infections appeared in the elderly patients. These important aspects of our study have now been included in our revised manuscript. (Page 8 lines 131-133 and page 12 lines 216-219):

“From the date of HUNT entry and up until February 2017, 15,496 (22%) of the 79,393 participants were hospitalized due to a bacterial infection at least once. Background characteristics of our study populations are described in Table 1.”

“Our long follow-up time which naturally led to our participants being followed into their old years combined with the fact that most of the infections appeared in the elderly population, probably is the explanation why as many as 22% of the study participants had a hospital stay with infection during the follow-up time.”

Comment 6 

The sex distribution would be of interest especially with UTI at different ages. This was common in young adults – were they mostly women? 

Response: We agree that the sex distribution of the different foci of infection is of interest and have now included our table 1 divided into 2 tables (male and female) as supplementary tables 3 and 4. As suspected by the reviewer, the incidence rate of hospitalization due to UTI was higher in women. Interestingly, the 30-day mortality rate of UTI was higher in men, which was also the case for pneumonia and sepsis/bacteraemia. This has been addressed in the revised manuscript on page 10, lines 173-175:

“This again increased steeply with age from 90 at the age of 30 to 2473 after the age of 80, and there was a marked difference between men and women, with women having an incidence rate of 637 per 100,000 (S3 and S4 Tables).”

Comment 7 

Pre morbidities – did they play a factor in the incidence of infections. Fragility and low immunity in older population was discussed – but what of underlying illnesses? 

Response: The role of pre morbidities is certainly an interesting factor here, both underlying medical diagnoses and modifiable lifestyle factors. We judged this to be out of the scope of this article, however we will in future articles try to address this. These thoughts are now included in the revised manuscript on page 13, lines 226-228:

“In addition to ageing, the role of other underlying conditions and modifiable risk factors is certainly interesting factors in future research.”

Comment 8 

Any idea what organisms were cultured in the different infections and age ranges – this would be useful information to guide clinical care. 

Response: Yes, we do have the identity of all the different organisms in the positive blood cultures. We clearly agree that information on types of microbes and their resistance patterns stratified by different infection focuses, sex and age groups, is of importance for guiding clinical practice. In fact, we have plans to include such information in a forthcoming paper focusing in å broader approach on individual risk for infections. Accordingly, we have chosen not to include these data here, but if the reviewer has a strong opinion on this, we are willing to do additional analyses on this in the current manuscript. 

Comment 9 

The study was done in Norway – but most comparisons in discussion were with USA studies. Any related research in Europe – if so ow comparable are the findings? 

Response: We have tried to include European studies as well, to make sure the results are as comparable as possible. Studies from France, Sweden, Denmark and Norway have been included and discussed but we do agree that there is an overweight of North-American studies. This has now been commented on the revised manuscript on page 17 lines 335-337:

“We have compared our results with earlier results from different countries, however an overweight of comparable studies were from USA. It is uncertain how comparable our cohort is to an American population.”

Comment 10 

Was there some temporal variations with the infections esp. related to weather changes. 

Response: The reviewer raises an important aspect of epidemiology of infectious diseases. Norway is country where the weather and temperature varies substantially throughout the year. A table showing incidence rates of the four main foci of infection divided into autumn/winter and spring/summer is now included in the supplements section, and the main findings here are discussed in the revised manuscript on page10, lines 159-161:

“Pneumonia was the focus of infection which seemed to have the largest seasonal difference, with a markedly higher incidence rate in September- February compared to the warmer months. (S2 Table).”

he

Comment 11 

Minor typos – Sepsis results section line167 UTI not UVI, and line 168 a not av. 

Response: Thank you, this has now been corrected. 

Comment 12 

Overall, the strength of this study is a large population cohort. Some biases to be considered could be changes within the health care systems over the years, as well as changes in health seeking behavior in a more informed cohort over time

Response: Thank you for highlighting these two important aspects. These 2 mentioned biases have now been taken into the revised text on page 17, lines 326-328.

“Our long-time follow up is a strength, but also brings the challenge of having to consider changes within the healthcare system and changes in health-seeking behaviour in a more informed cohort over time.”

Reviewer 2:

This study gives a population-based overview of common severe infections and sepsis in a Norwegian area with a background population of 130,000. The study is mainly descriptive. I have some major concerns that confuse me:

Comment 1

It is based on patients that have agreed to participate in a study, based on questionnaires, clinical examination, and retrieval of blood collection. However, this is mentioned once and then all the data used in the study are registry-based. Does the legislation in Norway not allow retrieval of registry-based data without the consent of each individual patient? If so, this should be stated explicitly.

Response: We thank the reviewer for raising this issue and we understand that the previous wording may have been confusing. Our work uses data from two types of sources: 1) The HUNT Study (a series of cross-sectional surveys including biological sampling and physical examination), and 2) registry data (e.g. hospital diagnosis codes and date of death). The former requires consent from each individual, while the latter does not. To make this more clear, we have made some changes in the abstract (page 3, lines 40-43), and in the text made the following changes on page 6, lines 84-91 and page 7, lines 110-116:

“We used data from the second and third surveys, HUNT2 (1995-1997) and HUNT3 (2006-2008), respectively, in which a total of 79,393 subjects agreed to participate (69.5% and 54.1% of the invited population for HUNT2 and HUNT3). The majority of the participants (72% of the women and 69% of the men) in HUNT2 also participated in HUNT3. The participants completed questionnaires covering a wide range of health-related topics, underwent clinical examination and blood collection and were then followed from the day of first inclusion and up until February 2017. For all participants, we retrieved information on all hospital admissions to the county hospitals or the regional tertiary care hospital.”

“We retrieved the ICD-9 and ICD-10 codes for all hospitalizations of the study subjects in the county hospitals and to the regional tertiary care hospital. All Norwegian citizens are assigned a unique identification number at birth, and this number is registered in health care contacts. In addition to accessing the ICD codes upon discharge, this identification number was used to link data from the HUNT Study with the Norwegian population registry to obtain information on date of emigration and date of death, as well as to the hospitals´ information on positive blood cultures through February 2017.”

Comment 2

What is meant by “blood collection”? Does that include both biochemistry and microbiological specimens? If it also includes the latter, did that have any impact on the rate of positive blood cultures (i.e., not taken on indication only)?

Response: Blood collection upon agreeing to participate in the HUNT study consisted of different biochemical test with focus of being risk factors of later diseases (such as serum creatinine and serum ALAT). Blood cultures (and other microbiological tests) were not included in the tests obtained upon inclusion in HUNT, but were taken on clinical indication for hospitalized patients. We fully agree that this could have been explained better and this has been pointed out in the revised manuscript on page 7, lines 121-122:

“The blood cultures were taken on clinical indication only.”

Comment 3

The authors claim that this study is prospective. In what way is it prospective, especially given that data from the questionnaires apparently were not reported in this study? It seems to be a historic registry-based study.

Response The study is prospective as the patients are included on the date when they agreed to participate in the HUNT cohort, and are then followed over time, until they emigrated or died. Information was gathered from all hospital admissions during the follow-up time. This complete information on time-to-event from entry into the study allowed us to calculate incidence rates and risks. 

Comment 4

The time frames are confusing. The study mentions a 22-y time period, but later we find out that this includes two separate cohorts, one from 1995-1997, the other from 2006-8. Moreover, the expression “22-y follow up” is used frequently. The follow-up period differs between the two cohorts of which the oldest has around 11 years more to get recurrent infections etc. I cannot figure out where a 22-y follow-up period comes from. A more appropriate used of follow-up would e.g. be for the 30-d mortality rate, i.e. that is a 30-day follow-up period. A study period is not the same as a follow-up period.

An example: “From the date of HUNT entry and up until February 2017, the 79,393 participants had 37,298 hospital admissions with a bacterial infection (first-time and recurring event) (Fig 1).”: These participants were found in two cohorts (how many in each?), of which the first had 11 years more follow-up time. And if the age distribution was the same in the two cohort, those from the first cohort were 11 years older at the same calendar time, which has a huge impact on incidence etc. So these data are blurred and muddled. 

Response: We do agree that the time frames could have been explained clearer. In the first survey – HUNT2 – 65,665 participants were recruited between 1995 and 1997. As we had complete data on emigration, date of death and hospital admissions (and blood cultures) through February 2017, the first study subjects to enter HUNT2 were followed for up to 22 years. In 2006-2008, a new survey was conducted – HUNT3 – where 50,807 participants were evaluated, of which 13,728 were new subjects (i.e. had not participated in HUNT 2). As the reviewer points out. these last 13,728 participants were followed for up to 11 years. The median follow-up-time was 20.0 years (25th percentile 9.5 and 75th percentile 20.8). The varying time-at-risk has of course been accounted for in the incidence rate-calculations and the participants have not been counted twice if they joined both cohorts. 

A more thorough explanation is now included in the revised manuscript on page 6, lines 84-87 and page 8 lines 138-139:

We used data from the second and third surveys, HUNT2 (1995-1997) and HUNT3 (2006-2008), respectively, in which a total of 79,393 subjects agreed to participate (69.5% and 54.1% of the invited population for HUNT2 and HUNT3). The majority of the participants (72% of the women and 69% of the men) in HUNT2 also participated in HUNT3.

“The median follow-up-time was 20.0 years (25th percentile 9.5 - 75th percentile 20.8). ”

Comment 5

Nothing is mentioned about differences and similarities between the two cohorts. Were there any changes in incidence, mortality etc.??

Response: This is a very valid point made by the reviewer. Our reference number 6 explains and compares the different HUNT cohorts, but in the previous version of our manuscript we did not go into detail about these differences. As mentioned previously 72% percent of the women and 69% of the men that participated in HUNT2 also participated in HUNT 3. Overall, both HUNT2 and HUNT3 were representative of the adult Norwegian population. A table with background characteristics of the HUNT2 and HUNT3 population is now included. The original table 1 is renamed “Summary of results divided into eight different foci of infection” to avoid confusing it with the new table 2. 

Comment 6

It is difficult for non-Scandinavians to know how data from different registries were merged, as most countries do not have a unique personal identification number for their citizens.

Response: We agree that this is important to clarify. Accordingly, we have described how the unique personal identification number of Norwegian citizens was used to link the study population to all prospectively recorded blood cultures in the catchment area. We have pointed it out on page 7, lines 111-116:

“All Norwegian citizens are assigned a unique identification number at birth, and this number is registered in health care contacts. In addition to accessing the ICD codes upon discharge, this identification number was used to link data from the HUNT Study with the Norwegian population registry to obtain information on date of emigration and date of death, as well as to the hospitals´ information on positive blood cultures through February 2017.”

Comment 7 

Although the authors briefly describe the difficulties of defining sepsis, including the coding of these, they have omitted some important studies that show that the incidence of sepsis, based on ICD-codes, may vary more than three-fold [1,2]. These reviews were, amongst others, based on three studies [3-5], one of which was from Norway, but even this study is not mentioned in this manuscript [4].

Response: As stated by the reviewer, these are five important studies explaining the difficulties of retrospectively defining sepsis by using ICD-codes. To clarify, the reviewer´s reference number 1 was not omitted, but is reference number 13 in our original manuscript. With regard to reference number 2, this was omitted in the process of shortening-down the manuscript, but we agree that it highlights an important aspect in this discussion, and it is now included in the revised manuscript as reference number 43. The reviewer´s reference number 3 was listed as reference 10 in our original manuscript. We agree that Dr Flaatten has made important contributions to describe the sepsis epidemiology in Norway, and accordingly we chose to include three later studies from his research group as references (Knoop et al, Nygård et al and Nygård et al, our references no 15, 30 and 35).

Comment 8

I am not an expert on all the infections described in the article, but concerning pneumomia there are studies that give a thorough overview, are population-based, and have a very high number of patients [6].

Response: We agree with the reviewer that this paper by Thomsen et al is important. We have now included this as reference no 18 in the revised manuscript. Our estimated higher incidence rate fits well into their conclusion that pneumonia incidence is rising, as our estimations are performed up to 14 years later. Our mortality rates are similar. This study differs from ours due to the fact that we have a longer follow-up time, and we also look at recurrent infections. Our comments regarding this are now included in the revised manuscript on page 12, lines 207-211:

“A Danish study from 2006 (6) report incidence rates that are lower than we have found, however their main conclusion was that pneumonia incidence is on the rise. They report an increase in hospitalized pneumonia from 288 to 442 per 100 000 person-years from 1994 to 2003, and we found the rate to be 639 up to 14 years later.”

Comment 9 

It is confusing that some analyses are based on the first-time and some on the last time occurrence. The baseline should be the same (first-time occurrence) and competing-risk analyses [7] should be incorporated.

Response: We are sorry that this important topic seem confusing. When it comes to incidence rates for diagnosis codes we assessed first time occurrences. For recurrent infections, we evaluated recurrences after the first-time occurrence. When it comes to positive blood cultures our thought was that using both first time and later occurrences as baseline would be more clinically meaningful, however we see that this is a possible source of confusion. We have therefore now changed this, and the proportion of positive blood cultures are now throughout the paper calculated from only the first-time infections, including figure 2. This, has, overall, made this proportion larger. 

As is discussed in the manuscript, mortality rates are based on the last infection: This will necessarily give a higher case-fatality rate compared to not including recurring infections in the denominator. However, we feel that counting deaths from a last time occurrence will give the most correct result, seeing that this study not only describes the first-time occurrences but also the recurrences. From all the first-time occurrences, this way of counting the mortality rate will tell us who died from this first-time infection or a subsequent infection with the same focus. We have tried to clarify this in the revised manuscript in the Results section. (page 13 lines 230-233):

“Our mortality rates were based on the last infection of each focus. This has necessarily given a higher case-fatality rate compared to having the first-time infections as the denominator, however, this describes deaths from both the first-time infections and the recurrences and we believe it has given a more correct description of the total burden.”

Regarding competing risk analysis, this is a very interesting aspect, but we have chosen not to study different risk factors for infectious diseases in depth in this paper. We feel that competing risk analysis would have been more relevant if we had aimed at causal analysis between exposure and outcome. 

Comment 10 

“To the best of our knowledge, this is the first time such a large population cohort have been studied with such a long follow-up, a population comparable to the population of the rest of Norway/western world”: I tend to disagree, as mentioned above for pneumonia, and much larger background populations than 130,000 have also been the basis for studies of e.g. bacteremia [8].

Response: Many earlier studies have retrospectively counted the different diagnosis groups within a catchment area. We will still say that the fact that this is a population cohort followed over a longer time-period (median follow-up-time 20 years) combined with the fact that it looks at eight different foci of infection and looks at recurrence and death, makes this different and more complete. We, however, agree with the reviewer that this statement may be too categorical. The median follow-up time has been pointed out in the revised manuscript and we have now modified the wording on page 16-17, lines 316-320:

“A few population cohorts have earlier looked at bacterial infections, however a median follow-up time of a 20 years, the opportunity to look at recurring events, the linkage to the positive blood cultures and lastly the fact that we look at all the different foci of bacterial infection, gives us a unique insight into the total burden of disease in this population comparable to the rest of the western world. 

” 

Minor comments:

Comment 11

In the abstract is written: “Thorough background information on the total burden and severity of the different foci of infection will contribute to reduce this.”: How will some information per se reduce some burden and severity of different foci of infection??

Response: This is an interesting point made by the reviewer. The nature of infectious diseases is that they are, in large, preventable, i.e. the burden and severity can be reduced by preventative measures. Background information regarding this, in addition to information on the high risk of recurrence, will help in the decision-making regarding this on several aspects. Some examples are future research, future funding, focus on prevention of infections such as by vaccination (e.g. pneumococcal vaccine) and focus on identifying and treating different important modifiable risk factors. The abstract is now rewritten slightly in order to make this clearer. 

Comment 12 

Too many results from the tables and figures are reiterated in the text in the “Results” section.

Response: We appreciate the comment. Some of the results are now deleted from the text and are only in the table. 

References

1. Wilhelms SB, Huss FR, Granath G, et al. Assessment of incidence of severe sepsis in Sweden using different ways of abstracting International Classification of Diseases codes: difficulties with methods and interpretation of results. Critical care medicine. 2010 Jun;38(6):1442-9.

2. Gaieski DF, Edwards JM, Kallan MJ, et al. Benchmarking the incidence and mortality of severe sepsis in the United States. Critical care medicine. 2013 May;41(5):1167-74.

3. Angus DC, Linde-Zwirble WT, Lidicker J, et al. Epidemiology of severe sepsis in the United States: analysis of incidence, outcome, and associated costs of care. Critical care medicine. 2001 Jul;29(7):1303-10.

4. Flaatten H. Epidemiology of sepsis in Norway in 1999. Crit Care. 2004 Aug;8(4):R180-4.

5. Martin GS, Mannino DM, Eaton S, et al. The epidemiology of sepsis in the United States from 1979 through 2000. The New England journal of medicine. 2003 Apr 17;348(16):1546-54.

6. Thomsen RW, Riis A, Nørgaard M, et al. Rising incidence and persistently high mortality of hospitalized pneumonia: a 10-year population-based study in Denmark. J Intern Med. 2006 Apr;259(4):410-7.

7. Fine JP, Gray RJ. A proportional hazards model for the subdistribution of a competing risk. Journal of the American Statistical Association. 1999;94:496-509.

8. Nielsen SL, Lassen AT, Gradel KO, et al. Bacteremia is associated with excess long-term mortality: A 12-year population-based cohort study. J Infect. 2015 Sep 9;70(2).

---

## [Decision Letter · Decision Letter 1]

9 May 2022

PONE-D-21-20080R1Incidence, recurring admissions and mortality of severe bacterial infections and sepsis over a 22-year period in the population-based HUNT StudyPLOS ONE

Dear Dr. Liyanarachi,

Thank you for submitting your manuscript to PLOS ONE. After careful consideration, we feel that it has merit but does not fully meet PLOS ONE’s publication criteria as it currently stands. Therefore, we invite you to submit a revised version of the manuscript that addresses the points raised during the review process.

please revise following reviewers advice before acceptance.

We look forward to receiving your revised manuscript.

Kind regards,

Kazumichi Fujioka

Academic Editor

PLOS ONE

Journal Requirements:

Reviewers' comments:

Reviewer's Responses to Questions

**Comments to the Author**

1. If the authors have adequately addressed your comments raised in a previous round of review and you feel that this manuscript is now acceptable for publication, you may indicate that here to bypass the “Comments to the Author” section, enter your conflict of interest statement in the “Confidential to Editor” section, and submit your "Accept" recommendation.

Reviewer #1: (No Response)

Reviewer #2: (No Response)

2. Is the manuscript technically sound, and do the data support the conclusions?

Reviewer #1: Yes

Reviewer #2: Yes

3. Has the statistical analysis been performed appropriately and rigorously? 

Reviewer #1: N/A

Reviewer #2: Yes

4. Have the authors made all data underlying the findings in their manuscript fully available?

Reviewer #1: No

Reviewer #2: No

5. Is the manuscript presented in an intelligible fashion and written in standard English?

Reviewer #1: Yes

Reviewer #2: Yes

6. Review Comments to the Author

Reviewer #1: Authors have adequately addressed all the concerns from the first review appropriately. Some issues have been revised, some included in limitations while others re considered for potential follow-up work – which is acceptable.

One minor issue to consider is - whether patients with UTI died OF or died WITH UTI. This is more so since UTI was the second commonest cause of death following pneumonia. Were there any overlap infections at admission?

Much as one can clearly state some infections are a direct cause of death - I am not sure if one can say the same of UTI in most cases - as its common with very ill admitted patients esp. in the older population, - who may have another infection.

Comment 8 – regarding type of organisms need not be included in this manuscript.

Reviewer #2: I think the authors by and large have commented my questions and comments satisfactorily. It is, however, still a bit muddled with two cohorts and two follow-up periods although I have no doubt that incidence rates etc. have been computed correctly. I guess with the focus on age, the big differences between the two cohorts as regards median age (>10 y difference), death (28.9% vs. 3.9%), and follow-up time are okay.

There was one point that I think the authors have misunderstood (and admittedly, I did not explain it very clearly either!). They write:

“Regarding competing risk analysis, this is a very interesting aspect, but we have chosen not to study different risk factors for infectious diseases in depth in this paper. We feel that competing risk analysis would have been more relevant if we had aimed at causal analysis between exposure and outcome.”

When I referred to competing risk I did not think of different factors, but of death as competing risk to reinfection. That is, you can only be reinfected if you are alive. If e.g. a large proportion of a population dies shortly after the first infection, the denominator will change. It will actually change every time a person dies and it is therefore a good idea to incorporate a competing risk analysis, either using the Fine & Gray methods or treating death as censoring before time. Competing risk from death makes it hard to interpret your reinfection proportions. But again, as your study is mainly descriptive and not based on time-to-event regression analyses, I can live with that.

7. PLOS authors have the option to publish the peer review history of their article (what does this mean?). If published, this will include your full peer review and any attached files.

Reviewer #1: **Yes: **Harriet Mayanja-Kizza

Reviewer #2: **Yes: **Kim Oren Gradel

---

## [Author Response · Author response to Decision Letter 1]

3 Jun 2022

Comments from the reviewers:

Reviewer 1:

Authors have adequately addressed all the concerns from the first review appropriately. Some issues have been revised, some included in limitations while others re considered for potential follow-up work – which is acceptable.

One minor issue to consider is - whether patients with UTI died OF or died WITH UTI. This is more so since UTI was the second commonest cause of death following pneumonia. Were there any overlap infections at admission?

Much as one can clearly state some infections are a direct cause of death - I am not sure if one can say the same of UTI in most cases - as its common with very ill admitted patients esp. in the older population, - who may have another infection.

Comment 8 – regarding type of organisms need not be included in this manuscript.

Response: Thank you for highlighting this important aspect, which will always be important to consider when reporting the all-cause 30-day mortality following any event. We agree that this is especially important when it comes to the urinary tract infections, and it should have been highlighted. 

Of the 37298 participants having a hospital admission due to infection, 4628 had two infections codes and 918 had three or more. When reporting the 30-day all-cause mortality it is not possible to assess which of them contributed the most to death, and this is especially important to consider when it comes to the urinary tract infections. The manuscript has now been changed slightly on several places to make this important point clearer. (Lines 142-143, 160-162, 181, 188, 198-199 and 236-239).

Reviewer 2:

I think the authors by and large have commented my questions and comments satisfactorily. It is, however, still a bit muddled with two cohorts and two follow-up periods although I have no doubt that incidence rates etc. have been computed correctly. I guess with the focus on age, the big differences between the two cohorts as regards median age (>10 y difference), death (28.9% vs. 3.9%), and follow-up time are okay. 

There was one point that I think the authors have misunderstood (and admittedly, I did not explain it very clearly either!). They write: “Regarding competing risk analysis, this is a very interesting aspect, but we have chosen not to study different risk factors for infectious diseases in depth in this paper. We feel that competing risk analysis would have been more relevant if we had aimed at causal analysis between exposure and outcome.” When I referred to competing risk I did not think of different factors, but of death as competing risk to reinfection. That is, you can only be reinfected if you are alive. If e.g. a large proportion of a population dies shortly after the first infection, the denominator will change. It will actually change every time a person dies and it is therefore a good idea to incorporate a competing risk analysis, either using the Fine & Gray methods or treating death as censoring before time. Competing risk from death makes it hard to interpret your reinfection proportions. But again, as your study is mainly descriptive and not based on time-to-event regression analyses, I can live with that.

Response: We acknowledge the reviewer’s comment on competing risk of death by other causes than the severe infections studied. To assess the amount of competing risk of death with our commonest group of infection, pneumonia, as an example, we assessed the proportion of participants who were censored because of death 1) before their first pneumonia and 2) before any recurrent pneumonia. The corresponding numbers are 13131 and 4066.

When studying a population that ages during follow-up, death will serve as a competing risk. In our study we descriptively reported the proportion who were subsequently reinfected, which we think serves the purpose of informing about the burden of reinfections on the population level. However, in the original analyses patients that did not survive their first infection were included in the denominator when calculating the proportion of recurrence. In accordance with the reviewer’s comment, this has now been changed, leading to slightly higher proportions of recurrence in all diagnosis groups. This has been changed in Table 2 on page 9 and changed and explained on lines 50, 119-120, 159, 179, 184 and 197. Figure 2 has also been changed accordingly. For future in depth investigations of reinfections, including analyses of risk factors for reinfection, we agree that time-to-event analyses accounting for competing risk is an appropriate tool.

---

## [Editor Report · Decision Letter 2]

28 Jun 2022

Incidence, recurring admissions and mortality of severe bacterial infections and sepsis over a 22-year period in the population-based HUNT Study

PONE-D-21-20080R2

Dear Dr. Liyanarachi,

We’re pleased to inform you that your manuscript has been judged scientifically suitable for publication and will be formally accepted for publication once it meets all outstanding technical requirements.

Kind regards,

Kazumichi Fujioka

Academic Editor

PLOS ONE
---

## [Editor Report · Acceptance letter]

1 Jul 2022

PONE-D-21-20080R2 

Incidence, recurring admissions and mortality of severe bacterial infections and sepsis over a 22-year period in the population-based HUNT Study. 

Dear Dr. Liyanarachi:

I'm pleased to inform you that your manuscript has been deemed suitable for publication in PLOS ONE. Congratulations! Your manuscript is now with our production department. 

Kind regards, 

on behalf of

Dr. Kazumichi Fujioka 

Academic Editor

PLOS ONE